# Feasibility of a Cognitive-Behavioral-Therapy-Based Addiction Intervention for Relapse Prevention for Patients with Alcohol-Related Cognitive Impairments: A Controlled Pilot Study

**DOI:** 10.3390/jcm14207307

**Published:** 2025-10-16

**Authors:** Gwenny T. L. Janssen, Yvonne C. M. Rensen, Roy P. C. Kessels

**Affiliations:** 1Centre of Excellence for Korsakoff and Alcohol-Related Cognitive Disorders, Vincent van Gogh Institute for Psychiatry, 5803 DN Venray, The Netherlands; gwennyjanssen@vigogroep.nl (G.T.L.J.); yvonnerensen@vigogroep.nl (Y.C.M.R.); 2Department of Neuropsychology and Rehabilitation Psychology, Donders Institute for Brain, Cognition and Behaviour, Radboud University, 6525 GD Nijmegen, The Netherlands; 3Tactus Addiction Care, 7418 ET Deventer, The Netherlands

**Keywords:** alcohol-related disorders, neuropsychology, therapy, cognitive dysfunction, alcohol

## Abstract

**Background:** Cognitive disorders are highly prevalent in individuals with alcohol use disorder. Treatments have so far mainly focused on the amelioration of the cognitive impairments, but interventions to prevent relapse tailored to people with alcohol-related cognitive disorders are lacking. Here we present a new intervention aimed at people with alcohol-related cognitive disorders. **Methods:** In total, 59 inpatients with alcohol-related cognitive impairments participated in this study. A total of 37 completed the Relapse Prevention for patients with Alcohol-related Cognitive Disorder (RP-ACD) intervention and 22 received treatment as usual (TAU). The RP-ACD is a tailored group intervention for substance use disorder consisting of 12 one-hour group sessions. Outcome measures were the Alcohol Abstinence Self-Efficacy Measure (AASE-12), the Alcohol Urge Questionnaire (AUQ) and the Social Support Questionnaire (SSQ6). The overall experience was explored using a short in-house developed questionnaire. **Results:** Post-treatment, patients reported an improved self-efficacy compared to the pre-treatment baseline, but no differences were found on the other measures. No significant changes were found in the TAU group. Overall experiences and acceptability were rated positively. **Conclusions:** The RP-ACD intervention is a feasible and promising group-based addiction treatment for patients with alcohol-related cognitive impairment. A randomized and controlled study in a larger sample is required to establish its efficacy and effectiveness.

## 1. Introduction

Cognitive impairments are highly prevalent in people with chronic, excessive alcohol use and alcohol use disorder (AUD), with estimates of their prevalence varying between 30 and 80% [1]. All cognitive domains may be affected, and their severity may range from mild or moderate to major neurocognitive disorder [2]. These cognitive impairments are an important source of individual differences that affect many aspects of the addiction treatment [3]. That is, a patient’s inability to receive, encode, integrate, and employ intervention-related information may affect motivational processes and adherence, complicate long-term abstinence, affect the severity of relapse, and consequently hamper full recovery of the alcohol addiction [3,4]. Also, in general, people with AUD and other mental comorbidities—including neurocognitive disorders—have a higher risk of discontinuing their treatment [5]. In turn, there is evidence that prolonged abstinence from alcohol may promote cognitive recovery. Compared to early abstinence (i.e., up to 1 year), meta-analyses have shown that prolonged abstinence (after more than a year) results in less profound cognitive deficits [6], even though widespread cognitive deficits may still persist in the domains of executive function, processing speed, working memory, language and learning and memory [7]. Thus, preventing relapse in individuals at risk of developing alcohol-related cognitive impairments (ARCIs) is key, as it not only reduces the negative effects of the addiction itself, but also promotes cognitive recovery over time.

Interestingly, relapse prevention in individuals with ARCI has to date received little attention. Previous intervention studies in patients with ARCI have predominantly focused on cognitive training or remediation, aimed at the amelioration of the cognitive deficits rather than the alcohol use itself [8]. However, strong evidence regarding the efficacy of such cognitive remediation programs in individuals with ARCI is lacking [9]. Also, most studies on cognitive (re)training in people with cognitive impairments due to acquired brain injury show that the beneficial effects are usually limited to near-transfer improvement on specifically trained tasks, with little or no far transfer to everyday abilities [10]. Optimizing interventions to prevent relapse and reduce craving may consequently result in more beneficial cognitive effects in the long-term compared to such a cognitive training approach and also reduce the risk of further cognitive decline in case abstinence is not realized. However, to date, there is little research on addiction interventions specifically aimed at individuals with alcohol-related cognitive disorders [3], even though tailored relapse prevention programs likely enhance long-term abstinence [11]. Such an intervention would ideally tailor therapeutic sessions to individuals with cognitive impairments, for instance, by reducing group sizes, employing external aids to support knowledge transfer, and by environmental adaptations [12]. In the current pilot study, we aim to provide first evidence on the feasibility of a newly developed intervention for relapse prevention, the Relapse Prevention for patients with Alcohol-related Cognitive Disorders (RP-ACD), with the aim to reduce craving, increase self-efficacy and promote social support as a coping strategy. This intervention is based on cognitive-behavioral therapy, with specific modifications to tailor the program to patients with mild-to-moderate alcohol-related cognitive disorders.

## 2. Materials and Methods

### 2.1. Patients

All participants were inpatients of the Center of Excellence for Korsakoff and Alcohol-Related Cognitive Disorders of the Vincent van Gogh Institute for Psychiatry in Venray, the Netherlands. Eligibility for admittance to our clinic was determined using the criteria outlined in Appendix A. At least 4 out of 12 had to be met to be indicative of alcohol-related cognitive impairment [13]. In addition, patients had to meet the criteria for severe alcohol use disorder, diagnosed in accordance with the DSM-5-TR criteria [14]. The Montreal Cognitive Assessment (MoCA) was administered for descriptive purposes [15]. Moreover, all patients underwent a thorough diagnostic work-up, consisting of an extensive neuropsychological assessment (consisting of intelligence testing, assessment of the cognitive domains of learning and memory, executive function, speed of information processing, social cognition, concentration, visuoconstruction, orientation, and language, and self-report and informant-based questionnaires, as well as performance validity testing), a neurological examination, standardized clinical observations, neuroimaging, and a review of the patients’ medical history by a multidisciplinary team to exclude other pathology.

A total of 59 patients participated in this study. The intervention was offered as part of the regular treatment program in the clinic and all patients gave permission for the use of the clinically obtained data for scientific research by written informed consent. Patients were at least 6 weeks abstinent from alcohol, verified via urinalysis, prior to the start of the intervention, as improvements in cognitive function have been reported to be most prominent in the first 6 weeks of abstinence [16]. A total of 37 patients enrolled in the RP-ACD group and 22 served as controls, receiving treatment as usual (TAU) in this period, as they were eligible for starting the intervention but could not enroll yet because of logistic reasons.

### 2.2. Intervention

The RP-ACD intervention is aimed to promote and increase appropriate coping skills, enhance self-efficacy in achieving alcohol abstinence, and reduce craving in patients with mild-to-moderate alcohol-related cognitive disorders. The program consists of 12 one-hour group sessions offered over a period of 6 weeks that encompassed psycho-education, coping skills training, and strategies to identify and avoid high-risk situations in daily life settings. The intervention is based on existing cognitive-behavioral therapy (CBT) guidelines for substance use disorders. CBT seeks to identify and modify the cognitive and behavioral processes that maintain addictive patterns. Interventions promote adaptive coping strategies and enhance self-regulatory capacities. Standard CBT protocols typically emphasize cognitive restructuring and behavioral change through functional analysis (i.e., examining the cause, context, and purpose of addiction), emotional regulation, and relapse prevention [17,18], while skill-based approaches prioritize behavioral modification and the development of alternative, intrinsically rewarding activities [19]. The current intervention integrates elements from both standard CBT protocols and skill-based approaches and is adjusted to account for neurocognitive impairments.

Modifications of the standard protocols include the selection of themes, adaptation of materials, exercises, and session structure. For instance, themed sessions are delivered with temporal separation rather than being consolidated into a single session. Information is presented concisely, utilizing simple language and concrete examples. Exercises are predominantly designed based on participants’ prior experiences. Both digital and physical visual aids (e.g., images, graphics) are incorporated, and participants are encouraged to actively engage throughout each session. Training sessions commence and conclude with a review of the session’s content. All sessions follow the same structure, which consists of psycho-education, role-playing exercises, exchange of personal experiences and homework assignments, and each session has a central topic that is discussed (see Appendix B for a detailed description). Groups consisted of 3 to 6 patients.

TAU consisted of a structured alcohol-free inpatient setting with interventions supporting a healthy lifestyle, exercise, and daily activities, various forms of occupational therapy, and individual sessions with one’s therapist. Allocation to the RP-ACD or control group was primarily based on availability and (care-)capacity.

### 2.3. Outcome Measures

A controlled pre–post pilot-study design was applied to evaluate the intervention’s feasibility and to provide first evidence supporting its efficacy with respect to self-efficacy, craving, and perceived social support.

Self-efficacy was measured with the 12-item version of the Alcohol Abstinence Self-Efficacy Measure (AASE-12) [20]. Here, patients had to rate how tempted they would be to use alcohol and how confident they would be to abstain from using alcohol in six situations on a 5-point Likert scale. The composite self-efficacy score is the difference score of confidence minus temptation, ranging from −30 to +30, with higher scores reflecting more self-efficacy).

Craving was assessed with the Alcohol Urge Questionnaire (AUQ) [21], which consists of 8 statements related to feelings and thoughts about drinking that have to be rated on a 7-point Likert scale. The AUQ total score is the sum of the individual item score (range 8–56) with higher scores reflecting more craving.

Perceived social support was measured with the Social Support Questionnaire—Short Form (SSQ6) [22]. This questionnaire consists of 6 items related to different aspects of social support (e.g., who can make you feel relaxed, on whom can you count when you need help, who comforts you when you are upset). For each item, the patient is asked to indicate the names of 1 to 9 supportive persons and to rate the overall satisfaction with that support on a 6-point Likert scale. The SSQ6 number score is the sum of the number of people to whom the patient can turn in various situations (range 0–54), the SSQ6 satisfaction score is the total score of the Likert ratings (range 6–36), expressing the level of satisfaction.

All self-report questionnaires were administered before start of the treatment and after treatment completion. A qualitative evaluation was performed using five open-ended questions (see Appendix C), administered in the last session of the RP-ACD intervention only.

The study design was pre-registered at Open Science Framework (OSF) (https://doi.org/10.17605/OSF.IO/NJM7A, accessed on 15 October 2025). The reporting of this study followed the Transparent Reporting of Evaluations with Non-randomized Designs (TRENDs) statement [23].

### 2.4. Analyses

Analyses were performed using IBM SPSS 29.0. Baseline characteristics were compared for the two groups using the appropriate parametric (Student’s *t*) or non-parametric test (Mann–Whitney *U*, χ^2^). Only complete pre–post pairs were analyzed using two-tailed non-parametric paired-sample Wilcoxon signed rank tests for the pre- and post-assessments to test the hypothesis that the intervention would reduce craving, improve self-efficacy, and enhance social support in the RP-ACD group but not in the TAU group. Missing data were not imputed given the small sample size, the pilot study design, and the high probability that these were not missing at random. Outcome data are presented as medians. Effect sizes (*r*) for the Wilcoxon test were computed using the following formula: $r=Z/\sqrt{N}$. 95% Confidence intervals for the medians were computed using the bootstrapping procedure.

## 3. Results

Figure 1 shows the flowchart of the study, illustrating that all data points were not available for all participants (the main reason was that patients were already discharged before the T1 assessment was completed). The MoCA total score was slightly lower in the TAU group compared to the RP-ACD group (*t*(57) = 3.32, *p* = 0.002). The groups did not differ with respect to age (*t*(57) = 1.93, *p* = 0.06), education level (*U* = 306.0, *Z* = 1.40, *p* = 0.16), or sex distribution (χ^2^(1) = 0.028, *p* = 0.87). The mean time between T0 and T1 was shorter for the RP-ACD group than for the TAU group (*t*(12.8) = 2.58, *p* = 0.023) (Table 1).

Figure 2 and Table 2 show the results for the craving, self-efficacy, and social support outcome measures for the available data points for both groups. Analyses showed a significant improvement from baseline to post-intervention on the AASE-12 in the RP-ACD group (*Z* = −2.46, *p* = 0.014) but not the TAU group (*Z* = −0.26, *p* = 0.80). No statistically significant change was observed in craving (AUQ RP-ACD: *Z* = −0.71, *p* = 0.48; TAU: *Z* = −0.56, *p* = 0.57) or social support, neither in the number of people (RP-ACD *Z* = −0.09, *p* = 0.93; TAU: *Z* = −0.24, *p* = 0.81) nor in satisfaction (RP-ACD *Z* = −1.834, *p* = 0.18; TAU: *Z* = −0.18, *p* = 0.85). Visual qualitative inspection of the responses to the open-ended questions indicated that the intervention was evaluated positively: participants reported achieving their (personal) goals and were appreciative of the supporting workbook and SOS card as a means to maintain improvement in the nearby future.

## 4. Discussion

In this pilot study, we demonstrated that a newly developed intervention for relapse prevention in individuals with alcohol-related cognitive disorders (the Relapse Prevention for people with Alcohol-related Cognitive Disorders; RP-ACD), is feasible. Furthermore, using a controlled pre–post design, we showed that in the RP-ACD group, the patients’ abstinence self-efficacy improved (i.e., an increased confidence to abstain from using alcohol accompanied by a reduced temptation to use alcohol in specific situations), with a large effect size. This finding is very much in line with the content and aims of the intervention, which targets craving, peer pressure, slips, and relapses in every-day situations using goal setting, role-playing, and homework assignments. However, current craving and alcohol urge was not reduced after the intervention compared to the baseline assessment in the RP-ACD group. Visual inspection of the data shows that the pre-assessment alcohol urge was already relatively low, leaving little room for further reduction. However, it should be noted that while this assessment was conducted before the start of the RP-ACD intervention, patients were already abstinent from alcohol for at least 6 weeks, verified by urinalysis, and were inpatients of a ‘dry’ clinic with no access to alcohol, which may have substantially reduced physical craving symptoms compared to the levels pre-admission. Indeed, in untreated AUD populations, much higher AUQ scores have been reported (e.g., 28.8 in a community AUD population [25]). The number of people patients confided in did not change after the intervention compared to the baseline, neither did the level of social support satisfaction. It should, however, be noted that there was a substantial amount of missing data for the SSQ6 (especially on the satisfaction part), possibly reflecting the lack of a social support system, which is common in patients with AUD. Finally, the participants rated the intervention positively. No significant changes over time were observed in the TAU group.

Whilst these findings are promising, they are also preliminary in nature, as the present study should be regarded as a pilot study. That is, our study design comes with important limitations. First, it is not a randomized clinical trial. Allocation to either the RP-ACD or the TAU group was performed because of logistic reasons; as a result, the TAU group can be considered a waiting-list control group, as these patients would enroll in the RP-ACD treatment later on. Another limitation is the small sample size (especially for the TAU group) and the relatively large number of patients who did not complete all self-report measures post-intervention, notably in the TAU group, which may explain null-results. Moreover, the time between the baseline and post-interventions was longer for the TAU group than for the RP-ACD group and the lack of follow-up assessments on relapse is a limitation. Furthermore, we do not have detailed information on the patients’ somatic or psychiatric comorbidity or medication. A future trial could also include more context-sensitive outcome measures, such as coping skills, self-regulation tasks, or ecological momentary assessments of craving [26].

## 5. Conclusions

In conclusion, this controlled pilot study showed the feasibility of the RP-ACD intervention, a cognitive-behavioral-therapy-based addiction intervention tailored towards individuals with alcohol-related cognitive disorders. First results are promising: after the intervention, higher levels of self-efficacy were reported in the RP-ACD group but not in the TAU group. No reliable effects on drinking urge and social support were found. This null finding is possibly the result of our outcome measures not being sensitive to detect changes in a controlled inpatient setting with enforced abstinence. Participants rated the intervention and materials positively and found the intervention acceptable. The results of our non-randomized, small-sample pilot study suggest that the RP-ACD intervention is a promising group-based treatment to prevent relapse in alcohol use for patients with mild-to-moderate alcohol-related cognitive impairments, which to date is lacking [13]. A larger, adequately powered randomized and controlled study with longer-term follow-up assessments and relapse outcomes is required to establish the definitive efficacy and clinical effectiveness of the RP-ACD.

## Figures and Tables

**Figure 1 jcm-14-07307-f001:**
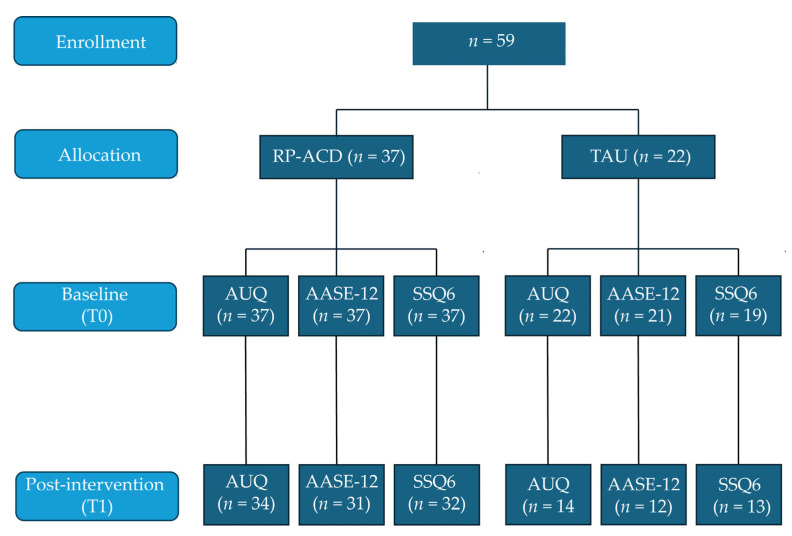
Flowchart of the pilot study. RP-ACD = Relapse Prevention for patients with Alcohol-related Cognitive Disorders (RP-ACD); TAU = treatment as usual; AUQ = Alcohol Urge Questionnaire; AASE-12 = Alcohol Abstinence Self-Efficacy Measure; SSQ6 = Social Support Questionnaire—Short Form.

**Figure 2 jcm-14-07307-f002:**
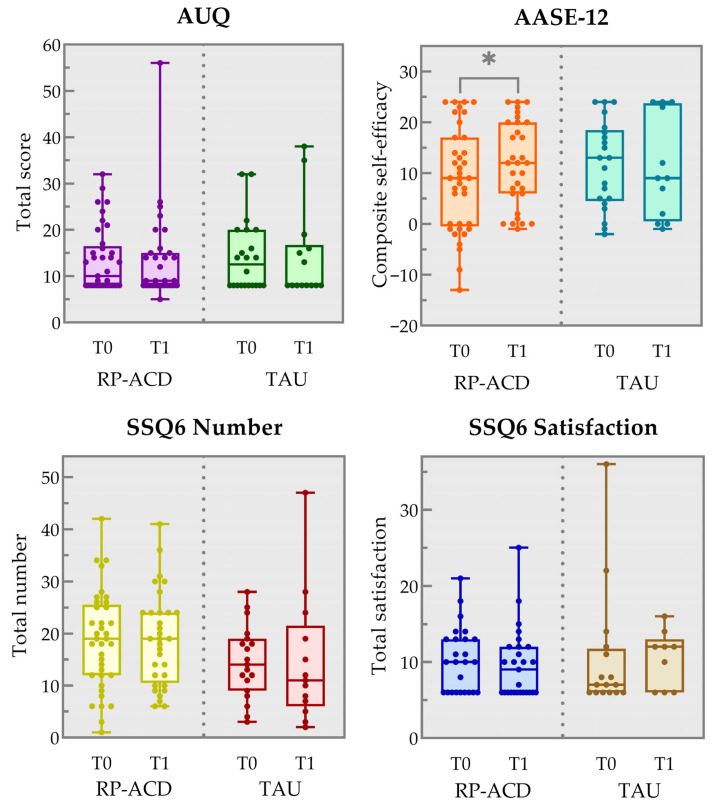
Box and whisker plots showing the results for the pre-intervention (T0) and post-intervention (T1) timepoints on the Alcohol Urge Questionnaire (AUQ total item score), the Alcohol Abstinence Self-Efficacy Measure (AASE-12 composite self-efficacy score) and the Social Support Questionnaire (SSQ6 total number of people and total satisfaction score) for the Relapse Prevention for patients with Alcohol-related Cognitive Disorders (RP-ACD) group and the treatment as usual (TAU) group. The boxes indicate the 25th and 75th percentile, the center line represents the median and the whiskers the minimum and maximum values. * *p* = 0.014 (two-tailed).

**Table 1 jcm-14-07307-t001:** Baseline characteristics for the Relapse Prevention for Patients with Alcohol-related Cognitive Disorder (RP-ACD) intervention group and the treatment as usual (TAU) group.

	RP-ACD Group	TAU Group
Total sample size (*n*)	37	22
Age	58.1 (9.5)	62.7 (7.8)
Sex (m:f)	26:11	15:7
MoCA Total Score	23.5 (2.6)	21.3 (2.4)
Education (*n*)		
Low	12	9
Average	13	12
High	10	1
Time between T0 and T1	39.5 (7.3)	57.7 (24.9)

Notes. Reported as mean (*SD*) unless otherwise specified. MoCA = Montreal Cognitive Assessment. Education was classified in accordance with the Dutch classification system that uses 7 levels of education [24] as low (levels 1–4; 9 years of education or less), average (level 5; 10–11 years of education), or high (levels 6–7; 12 years of education or more). Education level was missing for two patients.

**Table 2 jcm-14-07307-t002:** Median pre- and post-intervention scores on the outcome measures for the Relapse Prevention for patients with Alcohol-related Cognitive Disorders (RP-ACD) group and the treatment as usual (TAU) group, and change scores, effect sizes, and levels of significance.

	RP-ACD											TAU										
	T0			T1			Δ					T0			T1			Δ				
Outcome Measure	*n*	Median	95% CI	*n*	Median	95% CI	*n*	Median	95% CI	*r*	*p*	*n*	Median	95% CI	*n*	Median	95% CI	*n*	Median	95% CI	*r*	*p*
AUQ Total score	37	10.0	8.0, 14.0	34	9.0	8.0, 14.0	34	0.0	0.0, 3.5	0.12	0.48	22	12.5	8.0, 16.0	14	8.0	8.0, 16.0	14	0.0	−3.0, 3.0	0.15	0.57
AASE-12 Composite	37	9.0	6.0, 13.0	31	12.0	7.0, 17.0	31	3.0	1.0, 9.0	0.44	0.014 *	21	13.0	5.0, 17.0	12	9.0	1.0, 23.5	11	0.0	−5.0, 6.0	−0.08	0.80
SSQ6 Number	37	19.0	14.0, 22.0	32	19.0	14.0, 23.0	32	0.5	−3.5, 3.0	0.02	0.93	19	14.0	11.0, 18.0	13	11.0	7.0, 19.0	13	0.0	−5.0, 6.0	−0.07	0.81
SSQ6 Satisfaction	25	10.0	6.0, 13.0	27	9.0	6.0, 11.0	17	−1.0	−4.0, 0.0	−0.32	0.18	16	7.0	6.0, 11.0	9	12.0	6.0, 14.0	7	0.0	−2.0, 2.0	0.07	0.85

Abbreviations: 95% CI = 95% confidence interval. AUQ = Alcohol Urge Questionnaire; AASE-12 = Alcohol Abstinence Self-Efficacy Measure; SSQ6 = Social Support Questionnaire—Short Form. Note that a positive Δ and effect size (*r*) represent improvements (i.e., less craving, higher self-efficacy and more social support). * *p* < 0.05.

## Data Availability

The data underlying this article cannot be shared publicly as the informed consent forms did not include the possibility for public data sharing in a repository. The data can be obtained upon reasonable request from the corresponding author.

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
