# Peer review of "Feasibility of a Cognitive-Behavioral-Therapy-Based Addiction Intervention for Relapse Prevention for Patients with Alcohol-Related Cognitive Impairments: A Controlled Pilot Study"

_jcm, 2025, doi:10.3390/jcm14207307_

Round 1

Reviewer 1 Report

Comments and Suggestions for Authors

This manuscript presents a well-conceived pilot study evaluating the feasibility of a CBT-based relapse-prevention program (RP-ACD) tailored for individuals with alcohol-related cognitive impairment (ARCI). The study addresses a critical gap in addiction treatment research, as relapse-prevention strategies adapted to cognitive deficits are rare. The intervention is carefully designed with appropriate modifications to accommodate cognitive limitations and is clearly described.

The results are consistent with the intervention’s design. Patient-reported acceptability adds strength to the feasibility claim. Limitations (non-randomized design, small sample, missing data, lack of relapse outcomes) are acknowledged by the authors, and they appropriately frame the study as preliminary.

Overall, the article makes a meaningful contribution to the field of addiction treatment, particularly for cognitively impaired populations.

Author Response

Comment #1: This manuscript presents a well-conceived pilot study evaluating the feasibility of a CBT-based relapse-prevention program (RP-ACD) tailored for individuals with alcohol-related cognitive impairment (ARCI). The study addresses a critical gap in addiction treatment research, as relapse-prevention strategies adapted to cognitive deficits are rare. The intervention is carefully designed with appropriate modifications to accommodate cognitive limitations and is clearly described.

The results are consistent with the intervention’s design. Patient-reported acceptability adds strength to the feasibility claim. Limitations (non-randomized design, small sample, missing data, lack of relapse outcomes) are acknowledged by the authors, and they appropriately frame the study as preliminary.

Overall, the article makes a meaningful contribution to the field of addiction treatment, particularly for cognitively impaired populations.

Response #1: We thank the reviewer for the very positive remarks.

Reviewer 2 Report

Comments and Suggestions for Authors

  • Please clarify how the study adheres to recognized reporting guidelines for pilot/feasibility studies, such as the CONSORT extension for pilot trials or the TREND statement for non-randomized designs. Explicitly stating this will help readers interpret feasibility outcomes appropriately.

  • Provide more details on baseline comparability between groups. A table of participant characteristics (age, sex, education, baseline MoCA, AUD severity, comorbidities, medication use, etc.) with statistical comparisons would strengthen confidence that observed differences were not driven by imbalances.

  • Specify the eligibility thresholds clearly: for example, what MoCA cutoff was applied to define cognitive impairment? Was it ≤25, ≤23, or another criterion? Also clarify if participants just above the threshold were excluded.

  • The manuscript mentions an “extensive neuropsychological assessment.” Please describe the battery of tests used (e.g., memory, attention, executive function measures), or at least cite a standard protocol. This detail is essential to evaluate the validity of the cognitive screening process.

  • Report the sample size per outcome at each timepoint (T0 and T1), separately for intervention and control groups. Present these numbers in the flowchart and in the results tables.

  • Describe how missing data were handled. Were analyses conducted on a complete-case basis, or were missing data imputed? If imputation was not performed, please justify and discuss potential bias.

  • Report effect sizes (e.g., Wilcoxon r, Cohen’s d for nonparametric data) along with 95% confidence intervals. This will allow readers to judge the magnitude and precision of the observed changes, beyond p-values.

  • Present median (IQR) values at T0 and T1 for each group, along with change scores (Δ). This information should be tabulated, as it is difficult to extract from the current figure alone.

  • The choice of craving and social support as secondary outcomes requires stronger justification. Given that participants were in a controlled inpatient setting with several weeks of enforced abstinence, craving levels were already low and unlikely to change substantially. Please explain why these outcomes were prioritized and acknowledge their limited sensitivity in this context.

  • Consider discussing alternative or more context-sensitive outcomes (e.g., coping skills, self-regulation tasks, or ecological momentary assessments of craving) that might be used in future trials.

  • Temper the interpretation of these null findings: rather than framing them as “no effect,” emphasize that the outcomes were not well-suited to detecting change in the inpatient abstinent population.

  • In Figure 2, correct the terminology: AUQ stands for Alcohol Urge Questionnaire, not “Alcohol Use Questionnaire.”

  • Add a results table that provides median, IQR, and change scores per group (intervention vs control), alongside effect sizes and 95% CIs. This will make the findings clearer than relying on plots alone.

  • Ensure all figure captions are complete and accurate, with measures correctly identified and sample sizes stated for each analysis.

  • If space permits, include a baseline characteristics table to supplement the CONSORT-style flow diagram.

  • Reframe the conclusions to emphasize that the study primarily demonstrates feasibility and acceptability, not definitive efficacy. For example, highlight that participants found the intervention acceptable, retention was adequate, and the primary outcome (self-efficacy) showed promising improvement.

  • Avoid overgeneralizing efficacy claims, since the study was non-randomized, small-sample, and lacked follow-up.

  • Recommend that future work should test the intervention in a randomized, adequately powered trial with longer-term follow-up and relapse outcomes to determine true clinical effectiveness.

Comments on the Quality of English Language

The manuscript is generally understandable, but the English could be polished for clarity and flow. Some sentences are lengthy and could be simplified, and minor grammatical and typographical errors should be corrected. Careful editing will improve readability and ensure that the key findings are communicated more clearly.

Author Response

Comment #1: Please clarify how the study adheres to recognized reporting guidelines for pilot/feasibility studies, such as the CONSORT extension for pilot trials or the TREND statement for non-randomized designs. Explicitly stating this will help readers interpret feasibility outcomes appropriately.

Response #1: We followed the Transparent Reporting of Evaluations with Nonrandomized Designs (TREND) statement, and a completed checklist was also uploaded with the manuscript. As this was not explicitly mentioned in the manuscript, we have added the following statement to the revised Methods section: “The reporting of this study followed the Transparent Reporting of Evaluations with Non-randomized Designs (TREND) statement” (lines 158-159)

Comment #2: Provide more details on baseline comparability between groups. A table of participant characteristics (age, sex, education, baseline MoCA, AUD severity, comorbidities, medication use, etc.) with statistical comparisons would strengthen confidence that observed differences were not driven by imbalances.

Response #2: We have added a table to the results section presenting the baseline variables for the two groups (lines 181-186). All participants met the criteria for severe alcohol use disorder. We do not have a quantification of the amount of drinking over the years, as these cannot be reliably assessed by self-report, especially in individuals with cognitive deficits. Our database does not include a list of medication used (which may vary over time) or co-morbid diseases or disorders, which are, however, common in our patient sample. We have mentioned this as a limitation the revised Discussion. We have also compared the two groups statistically with respect to the available baseline variables, showing a slightly higher MoCA score in the RP-ACD group and longer time between T0 and T2 in the TAU group. Since our study is not an RCT, baseline differences are likely to be present.

Comment #3: Specify the eligibility thresholds clearly: for example, what MoCA cutoff was applied to define cognitive impairment? Was it ≤25, ≤23, or another criterion? Also clarify if participants just above the threshold were excluded.

Response #3: We have rephrased the eligibility criteria. Admission to our clinic is based on a 12-item checklist, of which at least 4 criteria have to be answered positively to be indicative for alcohol-related cognitive disorder. The MoCA was administered in all patients, but the sensitivity and specificity of the published cut-off scores is not optimal, so we did not use this as an exclusion criterion (we have rephrased this in our revised manuscript, i.e., the MoCA was administered for descriptive purposes). See lines 77-83.

Comment #4: The manuscript mentions an “extensive neuropsychological assessment.” Please describe the battery of tests used (e.g., memory, attention, executive function measures), or at least cite a standard protocol. This detail is essential to evaluate the validity of the cognitive screening process.

Response #4: The diagnostic neuropsychological work-up in our clinic is extensive and includes intelligence testing (full Wechsler Adult Intelligence Scale-IV), assessment of all cognitive domains (memory: Location Learning Test-Revised, Rey Auditory Verbal Learning Test, Visual Association Test - Extended, Rivermead Behavioural Memory Test - Story Recall, Corsi Block-Tapping Test; executive function: Stroop Color-Word test, Trail Making Test, D-KEFS Twenty Questions Test, Brixton Spatial Anticipation Test, Zoo Map test, Key Search Test; language: Controlled Oral Word Test, Sydney Language Battery (optional); concentration: d2 Test; social cognition: Theory of Mind Test - Revised, Social Norms Questionnaire, Emotion Recognition Test; visuoconstruction: Rey Complex Figure Test; orientation: CST-20, speed of information processing: Stroop, TMT) and self-report questionnaires (Apathy Motivation Index, Observable Social Cognition Rating Scale, Coping in Stressful Situations, Symptom Checklist 90-R, Geriatric Depression Scale-15) and informant-based questionnaires (Nijmegen-Venray Confabulation Scale-Revised, Observable Social Cognition Rating Scale, Apathy Motivation Index), as well as performance validity (Test of Memory Malingering, VAT-E). We have rephrased the section on neuropsychological assessment in the revised manuscript: (lines 84-91) “…extensive neuropsychological assessment (consisting of intelligence testing, assessment of the cognitive domains learning and memory, executive function, speed of information processing, social cognition, concentration, visuoconstruction, orientation and language, and self-report and informant-based questionnaires, as well as performance validity testing)…”.  We did not list all individual tests in the manuscript itself, as not all are globally available and referencing them all would make the reference list too lengthy.

Comment #5: Report the sample size per outcome at each timepoint (T0 and T1), separately for intervention and control groups. Present these numbers in the flowchart and in the results tables.

Response #5: The flowchart was included to visually present the sample sizes per outcome measure per time point for the intervention and control groups. We initially thought about presenting these numbers in a table, but felt that visual presentation of the whole flow would be more insightful. At the request of this reviewer (see the following comments) we have now also added a table (Table 2) with the results, in which we also listed the sample sizes per time point per outcome measure (lines 193-198).

Comment #6: Describe how missing data were handled. Were analyses conducted on a complete-case basis, or were missing data imputed? If imputation was not performed, please justify and discuss potential bias.

Response #6: We have added a separate Analyses section to the revised version describing the analyses we performed in more detail (lines 161-172). Missing data were not imputed given the small sample size, the pilot study design, and the high probability that these were not missing at random. This is now explicitly stated in the analyses section. The issue of missing data is discussed in the Discussion section (lines 238-240).

Comment #7: Report effect sizes (e.g., Wilcoxon r, Cohen’s d for nonparametric data) along with 95% confidence intervals. This will allow readers to judge the magnitude and precision of the observed changes, beyond p-values.

Response #7: We have added Wilcoxon r effect sizes to the new table 2, as well as 95% confidence intervals for the medians reported in that table. These analyses are also described in the new analyses section (lines 161-172).

Comment #8: Present median (IQR) values at T0 and T1 for each group, along with change scores (Δ). This information should be tabulated, as it is difficult to extract from the current figure alone.

Response #8: We have added change scores to the revised table 2, as well as 95% confidence intervals for the medians (lines 193-198). Adding IQRs would make this table even more extensive and impossible to format in the journal template, and we feel this information does not add to the 95% CIs already presented (as well as the actual box and whisker plots in Figure 2).

Comment #9: The choice of craving and social support as secondary outcomes requires stronger justification. Given that participants were in a controlled inpatient setting with several weeks of enforced abstinence, craving levels were already low and unlikely to change substantially. Please explain why these outcomes were prioritized and acknowledge their limited sensitivity in this context.

Response #9: The inclusion of craving intensity and perceived social support as secondary outcome measures was guided by their established relevance in predicting relapse and long-term recovery of alcohol addiction, even in controlled inpatient settings. Although participants had achieved abstinence at the time of the intervention, craving has been shown to persist beyond acute withdrawal and may fluctuate in response to emotional states or environmental cues during treatment. Similarly, perceived social support is an established protective factor in addiction recovery and contributes to sustained abstinence. Furthermore, strengthening social support (i.e., coping strategies that facilitate help-seeking and the mobilization of social support) is a key element in the present intervention. We acknowledge that the sensitivity of these measures may be limited in this highly structured inpatient environment. We have added this to the Conclusion section: “No reliable effects on drinking urge and social support were found, a null finding that is possible the result of our outcome measures not sensitive to detect changes in a controlled inpatient setting with enforced abstinence.” (lines 262-264).

Comment #10: Consider discussing alternative or more context-sensitive outcomes (e.g., coping skills, self-regulation tasks, or ecological momentary assessments of craving) that might be used in future trials.

Response #10: We have added this to the Discussion section: “A future trial could also include more context-sensitive outcome measures, such as coping skills, self-regulation tasks, or ecological momentary assessments of craving [26].” (lines 254-256)

Comment #11: Temper the interpretation of these null findings: rather than framing them as “no effect,” emphasize that the outcomes were not well-suited to detecting change in the inpatient abstinent population.

Response #11: We have changed this sentence into: “No reliable effects on drinking urge and social support were found. This null finding is possibly the result of our outcome measures not being sensitive to detect changes in a controlled inpatient setting with enforced abstinence.” (lines 262-264)

Comment #12: In Figure 2, correct the terminology: AUQ stands for Alcohol Urge Questionnaire, not “Alcohol Use Questionnaire.”

Response #12: We apologies for the typo in the figure caption, which we have corrected (line 212).

Comment#13: Add a results table that provides median, IQR, and change scores per group (intervention vs control), alongside effect sizes and 95% CIs. This will make the findings clearer than relying on plots alone.

Response #13: This table has been added to the revised Results section (lines 193-198).

Comment #14: Ensure all figure captions are complete and accurate, with measures correctly identified and sample sizes stated for each analysis.

Response #14: The new table 2 provides all these details. Figure 2 is complementary and self-explanatory, as all data points are presented here. Adding sample sizes to figure 2 would compromise the visual presentation  (and these exact numbers are already presented in figure 1 and table 2) (lines 181-186)

Comment #15: If space permits, include a baseline characteristics table to supplement the CONSORT-style flow diagram.

Response #15: We have added a new table with all available baseline characteristics (Table 1).

Comment #16: Reframe the conclusions to emphasize that the study primarily demonstrates feasibility and acceptability, not definitive efficacy. For example, highlight that participants found the intervention acceptable, retention was adequate, and the primary outcome (self-efficacy) showed promising improvement.

Avoid overgeneralizing efficacy claims, since the study was non-randomized, small-sample, and lacked follow-up.

Recommend that future work should test the intervention in a randomized, adequately powered trial with longer-term follow-up and relapse outcomes to determine true clinical effectiveness.

Response #16: We fully agree that the results of our pilot study should be interpreted with caution. We have rephrased the concluding remarks to tone down any ‘definitive’ claims: “No reliable effects on drinking urge and social support were found. This null finding is possibly the result of our outcome measures not being sensitive to detect changes in a controlled inpatient setting with enforced abstinence. Participants rated the intervention and materials positively and found the intervention acceptable. The results of our non-randomized, small-sample pilot study suggest that the RP-ACD intervention is a promising group-based treatment to prevent relapse in alcohol use for patient with mild to moderate alcohol-related cognitive impairments, which to date is lacking [9]. A larger, adequately powered randomized and controlled study with longer-term follow-up assessments and re-lapse outcomes is required to establish the definitive efficacy and clinical effectiveness of the RP-ACD.” (lines 262-271)

Comment #17: Comments on the Quality of English Language: The manuscript is generally understandable, but the English could be polished for clarity and flow. Some sentences are lengthy and could be simplified, and minor grammatical and typographical errors should be corrected. Careful editing will improve readability and ensure that the key findings are communicated more clearly.

Response #17: We have carefully reviewed the manuscript for language oddities and style.

Reviewer 3 Report

Comments and Suggestions for Authors

The manuscript ‘Feasibility of a Cognitive-Behavioral Therapy Based Addiction Intervention for Relapse Prevention in Alcohol-Related Cognitive Impairments: A Controlled Pilot Study’ is interesting and can be of value in the field of alcohol addiction.

The authors aim to present a new intervention for people with alcohol-related cognitive disorders. The RP-ACD intervention was used and compared to TAU (usual treatment). Post-treatment patients reported an improved self-efficacy compared to the pre-treatment baseline, while no changes were observed during in the TAU group.

The authors conclude that the RP-ACD intervention is a feasible and promising group-based addiction treatment.

I have some concerns:

The introduction is short, however informative.

Materials and Methods: The Authors said that an extensive neuropsychological assessment was used. Could you provide more information about this, specifically which neuropsychological tests were used?

The authors want to present a new intervention method; however, more information about the innovation of the RP-ACD is needed. Please say more about the specificity of the RP-ACD intervention. How it differ from TAU? What are specific tasks focused on the cognitive problems?

And please say more about participants’ cognitive impairments. You just report the MoCA average score; however, say what kind of cognitive problems these persons have? Working memory? Attention problems? or other.

The outcome measures and analyses included self-efficacy, perceived social support, and the Alcohol Urge Questionnaire. There is no assessment of cognitive impairments.

Social support or self-efficacy are not cognitive functions. We do not know whether cognitive problems have changed or not. The title of the manuscript and the method seem to be focused on cognitive improvement. The authors do not report the data on improving cognitive functions.  So, it is unclear what the aim of an intervention is. What was the aim of the manuscript/study? imroving cognitive functions? or examine self-efficacy/social support?

Author Response

The manuscript ‘Feasibility of a Cognitive-Behavioral Therapy Based Addiction Intervention for Relapse Prevention in Alcohol-Related Cognitive Impairments: A Controlled Pilot Study’ is interesting and can be of value in the field of alcohol addiction.

The authors aim to present a new intervention for people with alcohol-related cognitive disorders. The RP-ACD intervention was used and compared to TAU (usual treatment). Post-treatment patients reported an improved self-efficacy compared to the pre-treatment baseline, while no changes were observed during in the TAU group.

The authors conclude that the RP-ACD intervention is a feasible and promising group-based addiction treatment.

I have some concerns:

Comment #1: The introduction is short, however informative.

Response #1: We have expanded the Introduction also at the request of reviewer #2 (lines 42-73).

Comment #2: Materials and Methods: The Authors said that an extensive neuropsychological assessment was used. Could you provide more information about this, specifically which neuropsychological tests were used?

Response #2: The diagnostic neuropsychological work-up in our clinic is extensive and includes intelligence testing (full Wechsler Adult Intelligence Scale-IV), assessment of all cognitive domains (memory: Location Learning Test-Revised, Rey Auditory Verbal Learning Test, Visual Association Test-Extended, Rivermead Behavioural Memory Test - Story Recall, Corsi Block-Tapping Test; executive function: Stroop Color-Word test, Trail Making Test, D-KEFS Twenty Questions Test, Brixton Spatial Anticipation Test, Zoo Map test, Key Search Test; language: Controlled Oral Word Test, Sydney Language Battery (optional); concentration: d2 Test; social cognition: Theory of Mind Test-Revised, Social Norms Questionnaire, Emotion Recognition Test; visuoconstruction: Rey Complex Figure Test; orientation: CST-20, speed of information processing: Stroop, TMT) and self-report questionnaires (Apathy Motivation Index, Observable Social Cognition Rating Scale, Coping in Stressful Situations, Symptom Checklist 90-R, Geriatric Depression Scale-15) and informant-based questionnaires (Nijmegen-Venray Confabulation Scale-Revised, Observable Social Cognition Rating Scale, Apathy Motivation Index), as well as performance validity (Test of Memory Malingering, VAT-E). We have rephrased the section on neuropsychological assessment in the revised manuscript: (lines 84-88) “…extensive neuropsychological assessment (consisting of intelligence testing, assessment of the cognitive domains learning and memory, executive function, speed of information processing, social cognition, concentration, visuoconstruction, orientation and language, and self-report and informant-based questionnaires, as well as performance validity testing)…”.  We did not list all individual tests in the manuscript itself, as not all are globally available and referencing them all would make the reference list too lengthy.

Comment #3: The authors want to present a new intervention method; however, more information about the innovation of the RP-ACD is needed. Please say more about the specificity of the RP-ACD intervention. How it differ from TAU? What are specific tasks focused on the cognitive problems?

Response #3: We apologize that it was not fully clear that the RP-ACD intervention is an alcohol relapse prevention treatment, not a treatment of the cognitive disorder aimed to improve the cognitive functions themselves. We now address this in more detail in the revised introduction. The modifications of existing protocols are outlined in section 3.2 Intervention: (lines 116-127) “Modifications include the selection of themes, adaptation of materials, exercises, and session structure. For instance, themed sessions are delivered with temporal separation rather than being consolidated into a single session. Information is presented concisely, utilizing simple language and concrete examples. Exercises are predominantly designed based on participants' prior experiences. Both digital and physical visual aids (e.g., images, graphics) are incorporated, and participants are encouraged to actively engage throughout each session. Training sessions commence and conclude with a review of the session's content. All sessions follow the same structure, which consists of psychoeducation, role-playing exercises, exchange of personal experiences and homework assignments, and each session has a central topic that is discussed (see Appendix B for a detailed description). Groups consisted of 3 to 6 patients. The intervention was aimed at achieving and maintaining alcohol abstinence and maximizing self-efficacy.” Also we expanded the section on the RP-ACD intervention in the methods section under 3.2: (lines 106-115) “The intervention is based on existing cognitive behavioral therapy (CBT) guidelines for substance use disorders. CBT seeks to identify and modify the cognitive and behavioral processes that maintain addictive patterns. Interventions promote adaptive coping strategies and enhances self-regulatory capacities. Standard CBT protocols typically emphasize cognitive restructuring and behavioral change through functional analysis (i.e., examining the cause, context and purpose of addiction ), emotional regulation and relapse prevention [17; 18], while skill-based approaches prioritize behavioral modification and the development of alternative, intrinsically rewarding activities [19]. The current intervention integrates elements from both standard CBT protocols and skill-based approaches, and is adjusted to account for neurocognitive impairments”.

With respect to the TAU, no alcohol relapse intervention was provided, other than maintaining abstinence (see under 3.2 Intervention: (lines 128-131) “TAU consisted of a structured alcohol-free inpatient setting with interventions sup-porting a healthy life style, exercise and daily activities, various forms of occupational therapy and individual sessions with one’s therapist.”). Note that the existing CBT protocol for SUD is not the TAU arm of the present pilot study.

Comment #4: And please say more about participants’ cognitive impairments. You just report the MoCA average score; however, say what kind of cognitive problems these persons have? Working memory? Attention problems? or other.

Response #4: The cognitive profiles varied across participants, but were in line with what has been previously reported in individuals with severe AUD (also cited in meta-analyses references 6 and 7), i.e. deficits in speed of information processing, executive function, learning and memory and social cognition. The MoCA is not designed to present a reliable cognitive profile, therefore we only report the total score.  A detailed description of each participant’s cognitive test results is beyond the scope of the current manuscript, given that the cognitive performance itself is not a target or outcome measure of this intervention.

Comment #5: The outcome measures and analyses included self-efficacy, perceived social support, and the Alcohol Urge Questionnaire. There is no assessment of cognitive impairments.

Response #5: Improvement of cognitive function is not the aim of any alcohol relapse intervention. However, prolonged abstinence may promote cognitive recovery in the long term. We have added a section to the revised Introduction to highlight the aims of this intervention, its outcome measure, as well as possible (long-term) effects on cognition (i.e. either cognitive improvement or slowing down cognitive decline) (lines 60-63, 70-73).

Comment #6: Social support or self-efficacy are not cognitive functions. We do not know whether cognitive problems have changed or not. The title of the manuscript and the method seem to be focused on cognitive improvement. The authors do not report the data on improving cognitive functions.  So, it is unclear what the aim of an intervention is. What was the aim of the manuscript/study? improving cognitive functions? or examine self-efficacy/social support?

Response #6: We have modified the title and also description of the intervention to Relapse Prevention *for patients with* Alcohol-Related Cognitive Impairments to clarify that the treatment is performed in people with cognitive disorder, but that the aim of the intervention is alcohol relapse prevention. The study aims are now explicitly highlighted in the revised Introduction (see previous comments).

Round 2

Reviewer 2 Report

Comments and Suggestions for Authors

Well done